# The High-Throughput In Vitro CometChip Assay for the Analysis of Metal Oxide Nanomaterial Induced DNA Damage

**DOI:** 10.3390/nano12111844

**Published:** 2022-05-27

**Authors:** Andrey Boyadzhiev, Silvia Aidee Solorio-Rodriguez, Dongmei Wu, Mary-Luyza Avramescu, Pat Rasmussen, Sabina Halappanavar

**Affiliations:** Environmental Health Science and Research Bureau, Health Canada, Ottawa, ON K1A 0K9, Canada; andrey.boyadzhiev@hc-sc.gc.ca (A.B.); silvia.solorio-rodriguez@hc-sc.gc.ca (S.A.S.-R.); dongmei.wu@hc-sc.gc.ca (D.W.); mary-luyza.avramescu@hc-sc.gc.ca (M.-L.A.); pat.rasmussen@hc-sc.gc.ca (P.R.)

**Keywords:** benchmark concentration modelling, nanoparticles, genotoxicity, potency ranking, in vitro, zinc oxide nanoparticles, copper oxide nanoparticles, titanium dioxide nanoparticles, solubility

## Abstract

Metal oxide nanomaterials (MONMs) are among the most highly utilized classes of nanomaterials worldwide, though their potential to induce DNA damage in living organisms is known. High-throughput in vitro assays have the potential to greatly expedite analysis and understanding of MONM induced toxicity while minimizing the overall use of animals. In this study, the high-throughput CometChip assay was used to assess the in vitro genotoxic potential of pristine copper oxide (CuO), zinc oxide (ZnO), and titanium dioxide (TiO_2_) MONMs and microparticles (MPs), as well as five coated/surface-modified TiO_2_ NPs and zinc (II) chloride (ZnCl_2_) and copper (II) chloride (CuCl_2_) after 2–4 h of exposure. The CuO NPs, ZnO NPs and MPs, and ZnCl_2_ exposures induced dose- and time-dependent increases in DNA damage at both timepoints. TiO_2_ NPs surface coated with silica or silica–alumina and one pristine TiO_2_ NP of rutile crystal structure also induced subtle dose-dependent DNA damage. Concentration modelling at both post-exposure timepoints highlighted the contribution of the dissolved species to the response of ZnO, and the role of the nanoparticle fraction for CuO mediated genotoxicity, showing the differential impact that particle and dissolved fractions can have on genotoxicity induced by MONMs. The results imply that solubility alone may be insufficient to explain the biological behaviour of MONMs.

## 1. Introduction

Metal oxide (MO) nanomaterials (NMs) are among some of the most highly utilized engineered NMs worldwide, with an estimated global market share of USD 5.55 billion in 2019 and an anticipated 9.3% growth by 2027 [1]. MONMs have many varied applications such as food and cosmetic-grade pigments and additives, catalytic reagents, and as advanced electronics and complex materials fabrication components [2,3]. With increased utilization, manufacturing, and manipulation of MONMs, there are increased concerns for risks associated with exposure to these materials.

MONMs are known to induce a wide variety of effects in laboratory experimental models. Evidence for health impacts in humans comes from occupational studies involving mining activities, welding fumes and diesel exhaust [4,5,6,7,8,9]. Studies conducted in mammalian in vitro and in vivo model systems have shown pulmonary toxicity and DNA damage resulting from exposure to different kinds of MONMs [10], the exact nature and magnitude of which are suggested to be dependent upon their physical-chemical properties such as particle size, solubility and particle surface [3]. Recent research has shown that, in human volunteers, inhalation of zinc oxide (ZnO) nanoparticles (NPs) at concentrations under the US-recommended exposure limit for zinc fumes significantly increases markers of lung and systemic inflammation [11]. With respect to genotoxicity, NMs (including MONMs) have the potential to induce DNA damage through both primary and secondary mechanisms, which involve the direct interaction of NMs with DNA, the production of reactive intermediates such as reactive oxygen species (ROS) that can cause DNA damage, or through reactive molecules produced during an inflammatory response [12]. Moreover, some MONMs (e.g., titanium dioxide (TiO_2_) NPs) exhibit photoactivation. TiO_2_ NPs induce oxidative DNA damage in acellular conditions upon visible or UV irradiation [13]. Although early ROS production may be protective in nature, improperly repaired DNA damage arising from ROS can lead to mutagenic and carcinogenic effects. Three widely utilized and commercially available MONMs—ZnO, copper oxide (CuO) and TiO_2_—have been examined for their genotoxic potential using mammalian in vitro and in vivo model systems [14,15,16]. Both ZnO and CuO NPs are soluble in a biological medium [17,18] and their toxicity is thought to be a combination of both particulate and dissolved metal species. The microparticle (MPs) counterparts of CuO and ZnO NP types are shown to be less soluble [17,19] and differ in their toxicity potential compared to their nano forms. While both ZnO and CuO NPs have been consistently shown to induce DNA damage both in vitro and in vivo, there are conflicting reports with respect to TiO_2_ NPs [14]. With respect to TiO_2_ NPs, their insoluble nature precludes the impact of an ionic or dissolved fraction on response and any resulting genotoxicity is suggested to be the result of particle–cell interactions, or through the production of reactive intermediates; both of which are impacted by the material’s physical-chemical properties [14,20]. With respect to crystal phase and surface coating, most studies have evaluated the anatase form of TiO_2_ NPs and mixtures of anatase/rutile, while few studies have assessed the genotoxicity of coated TiO_2_ NPs [21]. Thus, a clear understanding of how size, solubility and other material properties affect the genotoxic potential of MONMs is still needed.

There are numerous tests available for measuring the genotoxicity of NMs in vitro, with varying levels of throughput and applicability. The most commonly used assays include the comet assay, the micronucleus assay, the chromosomal aberration test, and the hypoxanthine phosphorybosyl transferase forward mutation assay [22]. Of these, the comet assay has been the most widely utilized test for measuring DNA damage in eukaryotic cells [22,23,24]. This assay has been used to investigate the DNA damaging properties of NMs [25]. However, the traditional comet assay is low throughput, which makes it challenging for the purposes of screening a large number of NMs in a time- and resource-effective manner. Recent developments of this assay have yielded a commercially available high-throughput version, the CometChip assay [26], which holds great potential for routine screening of DNA damage induced by NMs.

The high-throughput alkaline CometChip assay relies on the same principles as the standard alkaline comet, however, it increases the throughput ~200 fold by utilizing a micro-patterned agarose chip in a 96-well format instead of glass slides [26]. Each of the 96-wells in the chip contains ~500 micropores, fitting one cell per pore, which allows for 96 different exposures simultaneously on one gel. This system has been used to successfully measure chemical-induced DNA damage in suspension and adherent mammalian cell cultures as well as in peripheral blood monocytes [26,27]. Similar in-house made micro-patterned agarose-gel comet chip systems have been used to assess the in vitro genotoxicity of ZnO, cerium dioxide (CeO_2_), iron oxide (Fe_2_O_3_), silica (SiO_2_), and silver (Ag) NPs as well as ZnO nanorods [28,29]. The high-throughput assay enables side-by-side comparison of all three species of metal oxides (NP, MP, and dissolved metal salt) for their DNA damaging potential and allows a systematic understanding of the contribution of different properties of MONMs to genotoxicity. This type of data is a prerequisite to applying approaches such as read-across in human health risk assessment of nanomaterials.

Thus, the overarching goal of the present study was to apply the high-throughput CometChip assay to investigate the role of size, solubility and surface coatings on NM-induced genotoxicity. Specifically, the study investigated DNA damaging effects in adherent murine lung epithelial (FE1) cells exposed for 2–4 h to different doses of ZnO, CuO, and TiO_2_ NPs, MPs as well as zinc and copper chloride salts. No-observed-effect-concentration (NOEC) and the lowest-observed-effect-concentration (LOEC) were used alongside benchmark concentration (BMC) modelling to determine points of departure (PODs) to delineate the differences in relative potency between the various exposures.

## 2. Materials and Methods

### 2.1. Metal Oxide Nanomaterials and Metal Chloride Salts

A total of 10 NPs with primary particle sizes of ≤ 50 nm, and one metal oxide nanowire (NW) were procured from five different manufacturers (Table 1). In addition, three pristine bulk analogues of TiO_2_, ZnO, and CuO (Table 1), copper (II) dihydrate (referred to as CuCl_2_; Sigma Aldrich, Catalogue #: C3279-100G), and zinc (II) chloride (ZnCl_2_; Sigma Aldrich, Catalogue #: Z0152-100G) were procured to assess the effects of particle size fraction and particle solubility on genotoxic response. A 30% hydrogen peroxide (H_2_O_2_) solution in dH_2_O (Sigma Aldrich, Catalogue #:216763) was used as a positive assay control for DNA damage.

### 2.2. Cell Culture

Immortalized FE1 cells derived from the Muta™ Mouse transgenic rodent model were utilized for particle and metal chloride exposures. These cells retain the characteristics of type I and type II pulmonary alveolar epithelial cells, and have been used in the past to assess the genotoxicity and mutagenicity of both chemicals and NPs [30,31,32,33,34,35].

Cells were maintained in phenol-red-containing Dulbecco’s Modified Eagle’s Medium Nutrient Mixture: F12 HAM (1:1) culture media (DMEM/F12 (1:1) (1X), Life Technologies, Burlington, ON, Canada) supplemented with 2% fetal bovine serum (FBS, Life Technologies, Burlington, ON, Canada), 1 ng/mL human epidermal growth factor (EGF, Life Technologies, Burlington, ON, Canada), 1 U/mL penicillin G, and 1 µg/mL streptomycin (Life Technologies, Burlington, ON, Canada) in an incubator at 37 °C, with 5% CO_2_. For CometChip exposures, the same conditions were used, except for the absence of phenol-red in the exposure media.

### 2.3. Primary Particle Size Determination

A JEM-2100F Field Emission transmission electron microscope (TEM) (JEOL, Peabody, MA, USA) was used to capture 9–10 non-overlapping transmission electron micrographs of dry ZnO and TiO_2_ MONMs used in this study for primary particle size analysis, as described in Boyadzhiev et al., 2021. All electron microscopy images were imported into ImageJ for particle size measurement. From the TEM images of the MONMs, the length and width of 9–18 individual particles per image were analyzed for a total of at least 100 particles per NM. For the TiO_2_ NW, only the width was measured due to the tangled nature of the material in the micrographs. The size distributions were plotted in histogram format (GraphPad Prism 9.2.0, GraphPad Software, San Diego, CA, USA), and the mean length and width were reported with standard deviation.

With respect to the metal oxide MPs, a JSM-7500F Field Emission scanning electron microscope (SEM) (JEOL, Peabody, St. Louis, MO, USA) was used to capture 10 non-overlapping scanning electron micrographs of the ZnO and TiO_2_ MPs used in this study. Due to the aggregated nature of the particles, it was not possible to conduct size analysis from the resulting micrographs. The CuO NPs and MPs have been similarly analysed for their primary particle size and results have been published [17].

### 2.4. Particle Dissolution Experiments

The release of Zn from the ZnO NPs and MPs was assessed as outlined in [17,36]. The ZnO NPs were suspended at a concentration of 1 mg/mL in ultrapure dH_2_O as described in [37]. The NP stock dispersions were sonicated at a delivered sonication energy of 265 J/mL. From this stock, serial dilutions in cell culture media +2% fetal bovine serum were made to final concentrations of 10 and 100 µg/mL. With respect to the ZnO MPs, the particles were directly suspended in cell culture media +2% fetal bovine serum at a concentration of 100 µg/mL and mixed by vortexing. ZnO particle suspensions were incubated on an orbital shaker (1 h shaking per day at 100 (rpm) rotations per minute shaking rate) within polypropylene 50 mL conical tubes at 37 °C for 0, 24, and 48 h. At the designated times, samples were withdrawn and the dissolved metal fraction was measured following sequential centrifugation (20,000× *g*, 3 × 30 min; Alegra 64R centrifuge, Beckman Coulter, Mississauga, ON, Canada). The final extracts were acidified to a concentration of 2.5% HNO_3_ (SCP Science, Graham Baie D’Urfé, QC, Canada) and the Zn fraction was measured using an Inductively Coupled Plasma Optical Emission Spectrophotometer (ICP-OES, Agilent Technologies, Santa Clara, CA, USA). Blank samples as well as spiked matrix blanks were incubated alongside the ZnO NP and MP suspensions. Analysis of the dissolved fraction was conducted using a 5100 Synchronous Vertical Dual View (SVDV) ICP-OES at a wavelength of 213.857 nm as recommended by the manufacturer. The instrument was operated at 1.2 kW power, 12 L/min plasma, 1 L/min auxiliary, 0.7 L/min nebulizer flow rate in SVDV mode, with 3 replicates per sample. The CuO NPs and MPs have been similarly assessed for their solubility and results have been published [17].

### 2.5. Preparation of Exposure Suspension

MONM and MP stock suspensions were prepared in ultrapure dH_2_O, based upon delivered sonication energies determined to produce a stable suspension for the NP in question. Sonication conditions for each particle can be found in Appendix A Appendix A. For all sonication procedures, a Branson Ultrasonics Sonifier™ 450 (Branson Ultrasonics Markham, ON, Canada) with a ½‣ disruptor horn and a removable flat tip was used. The tip was immersed into the suspension, 1–1.5 inches from the surface of the air–liquid interface. Rapid sedimentation of the TiO_2_ NPs, MK-TiO_2_-A050 (Table 1), was observed to occur in water. In order to improve the dispersion, bovine serum albumin (BSA; Sigma Aldrich, Oakville, ON, Canada) was used as a stabilizing agent (Appendix A Appendix A). NPs were suspended in ultrapure dH_2_O at 5 mg/mL. This suspension was sonicated as described above, immediately following which BSA was added to a final concentration of 2 mg/mL. The suspension was then vortexed and incubated for 10 min. Sonicated stock suspensions were used to produce desired serial dilutions in a cell culture medium. Each exposure dilution was inverted 20 times to mix and the dilutions were used within 15 min of preparation. Three to five concentrations of MONMs, MPs, CuCl_2_, and ZnCl_2_ were chosen for exposure. Dose interconversions for each are available in Appendix A Appendix A.

### 2.6. Dynamic Light Scattering Analysis of NP Suspensions

Particle characterization in the relevant medium was conducted using dynamic light scattering (DLS), as described in Boyadzhiev et al., 2021. Sonicated and suspended NP stocks in ultrapure dH_2_O were diluted in cell culture medium (+2% fetal bovine serum) to a final concentration of 50 μg/mL, and aliquots were used for DLS analysis in a Zetasizer Nano ZSP (Malvern Panalytical, Westborough, MA, USA). The hydrodynamic diameter (aggregate size measured in nm) and the poly-dispersity-index (PDI; a measure of the broadness of size distribution) were calculated. For each particle sample, 2–3 independent runs were conducted, with 5 measurements conducted per run for a total of 10–15 measurements per particle. For MK-TiO_2_-A050 TiO_2_ NPs, DLS characterization was conducted in DMEM (+ 2 % fetal bovine serum) and 0.02 mg/mL BSA.

### 2.7. Trypan Blue Exclusion Assay

In order to determine that the concentrations chosen for genotoxicity analysis do not induce overt cytotoxicity, the Trypan Blue exclusion method for cell viability assessment was utilized. FE1 cells were plated in 6-well plates at a density of ~130,000 cells/well. Following overnight incubation, the cells were exposed to 1.8 mL of 3–5 doses (0–108 µg/mL) of MONMs, MPs, and ZnCl_2_ or CuCl_2_ as outlined in Appendix A Appendix A. In all cases, blank media exposed cells served as negative controls. For MK-TiO_2_-A050 TiO_2_ NPs, negative controls were exposed to medium containing 0.04 mg/mL BSA (the amount of BSA that was present in the medium at 100 μg/mL NP concentration). Following 2 and 4 h of exposure, cell supernatant was removed and cells were washed once with 0.5 mL of PBS. Phase-contrast images of each condition were acquired at 4× and 20× in order to assess morphology. Cells were detached from the surface with 0.15 mL of 0.25% Trypsin-EDTA (1X) (Thermofisher Scientific, Whitby, ON, Canada), and resuspended in 0.5 mL of fresh culture medium. Trypan Blue staining was conducted as described in [17,32]. In brief, 10 μL of cell suspension was combined with 10 μL of Trypan Blue dye (Thermofisher Scientific, Whitby, ON, Canada) and incubated at room temperature for 5–10 min before counting in a hemocytometer. The number of blue and white cells was counted, and the ratio between the number of white cells and the total number of cells were used as a measure of cell viability. Each sample was assessed in triplicate (*n* = 3), and the statistical difference between the negative control (time-matched blank media samples) was determined through a one-way ANOVA with a Dunnett’s post hoc in the case of significant results in SigmaPlot 12.5 (Systat Software Inc., San Jose, CA, USA).

### 2.8. CometChip Assay

The protocol utilized for the CometChip experiments was a slight modification of the manufacturer protocol. On the day of exposure, FE1 cells were trypsinized and suspended at a density of ~150,000 cells/mL in fresh phenol red free growth medium. The suspension was passed through a 70 μm cell strainer (Fisher Scientific, Whitby, ON, Canada) in order to ensure that the suspension was single-celled. After 30 min of equilibration in 100 mL PBS (room temperature; Thermofisher Scientific, Whitby, ON, Canada), the micro-patterned agarose CometChip (Cedarlane Laboratories, Burlington, ON, Canada) was loaded into the macrowell former system, and the residual PBS was aspirated using a VacuSafe system (Integra LifeSciences, Toronto, ON, Canada). Next, 100 μL of the single-cell suspension was loaded into each well of the chip. The system was incubated at 37 °C for 15 min, after which the system was rocked east–west and north–south 4 times each in order to aid cell loading into the micropores and placed in the incubator for 5 additional minutes. Following loading, the cell-containing loading medium was aspirated from each well using a VacuSafe system and 50 μL of fresh phenol red free cell culture medium was loaded into each well. The system was placed back into the incubator until the exposure. For each experiment, a 4 h blank medium negative control was used and cells treated with 100 μM H_2_O_2_ in cell culture media for 1 h were used as positive assay controls. For MK-TiO_2_-A050, negative controls were exposed to medium containing 0.04 mg/mL BSA. Following exposure, the chip was washed with PBS and a low melting agarose (Cedarlane Laboratories, Burlington, ON, Canada) overlay was deposited onto the chip to fix the cells in place. The agarose was allowed to harden at room temperature before being transferred to a 4°C cold room to fully polymerise under light occlusion. The chip was then lysed for an hour in dark using 100 mL of the lysis buffer solution (Cedarlane Laboratories, Burlington, ON, Canada) and acclimatized in alkaline solution (pH ≥ 13, 200 mM NaOH (Sigma Aldrich, Oakville, ON, Canada) +1 mM EDTA (Thermofisher Scientific, Whitby, ON, Canada) +0.1% Triton-X (Fisher Scientific, Whitby, ON, Canada)) for 40 min. Electrophoresis was carried out in the dark in alkaline conditions (pH ≥ 13, 200 mM NaOH + 1 mM EDTA + 0.1% Triton-X), under constant voltage (20 V, 1 V/cm) and variable current (280 mA) for 50 min. The chip was neutralized in 400 mM and 20 mM Tris-HCl buffer (1M pH 7.4 Tris-HCl procured from Sigma Aldrich, Oakville, ON, Canada), followed by 14 h of overnight staining in dark in a solution of 0.2X SYBR GOLD stain (10,000× SYBR GOLD in DMSO procured from Life Technologies Corporation, Burlington, ON, Canada) in 20 mM Tris-HCl.

The following day, the chip was de-stained for one hour in 100 mL of room temperature 20 mM Tris-HCl and imaged using a Leica DMi8 automated confocal fluorescence microscope (Leica Microsystems, Wetzlar, Germany) at 5× magnification, and the resulting TIFF images were uploaded into the proprietary Trevigen Comet Software (Bio-Techne, Devens, MA, USA) for analysis. Quality control was conducted on the comets, in order to remove misidentified artefacts as well as to adjust improperly labelled comet tails and heads. Individual wells with less than 50 valid comets after quality control were not included in the final analysis. For all controls and experimental conditions, each biological replicate contained between 2–8 technical replicates. The mean percentage DNA in the tail was used as the metric for DNA damage. The final data, containing 3–4 biological replicates per condition were normalized via a log transformation and the resulting transformed data sets were statistically analyzed using a one-way analysis of variance (ANOVA) with a Dunnett’s multiple comparison post hoc against the respective media control in SigmaPlot 12.5 (Systat Software Inc., San Jose, CA, USA).

### 2.9. Benchmark Concentration Modelling

PROAST version 67.0 (https://www.rivm.nl/en/proast; accessed on 1 June 2020) was used to conduct concentration-response modelling of percentage DNA in the tail from FE1 cells exposed for 2–4 h. The benchmark response (BMR) was set to 1.0 (100% increase over baseline) according to the structure of the data and the magnitude of responses seen. BMC modelling was carried out (1) without using covariate analysis, and (2) by setting the exposure compounds as a covariate in PROAST, then carrying out modelling for the 2 and 4 h timepoints. All resulting models were assessed for goodness of fit, normality (through quantile-quantile plots), and homogeneity of variance (through residual plots). Outlier samples (samples with standardized residuals > 3) were removed for BMC modelling. Two dose metrics were used for BMC computation, µg/mL of the compound, and µg/mL of the constituent metal. All PROAST markdowns and model fitting graphs, quantile–quantile plots, and residuals plots can be found in the Appendix A.

## 3. Results

### 3.1. Primary Particle and Particle Suspension Characterization

All MONMs used in this study had a manufacturer reported primary particle size < 50 nm and specific surface areas between 33–65 m^2^/g. Regarding TiO_2_ NPs, the primary particle size was 20–50 nm for all except MKN-TiO_2_-A005, which was 8 nm in size with a surface area of 356 m^2^/g (Table 1). The MPs examined in this study had primary particle sizes ranging from 1000–5000 nm with specific surface areas between 2–8 m^2^/g (Table 2). SEM images were acquired for the ZnO and TiO_2_ MPs used in this study, however, the materials were too largely aggregated to conduct an accurate particle size analysis (Appendix A Appendix A). What can be seen from the representative micrographs is that the ZnO MPs have an irregular particle shape, with many rod-like crystallites present. The TiO_2_ MPs are largely spherical in appearance. Additional TEM imaging and primary particle size analysis of the NPs used in this study show that almost all NPs used in this study are roughly spherical, with measured sizes in the range of the manufacturer reported values (Table 1) (Appendix A Appendix A; CuO NP data previously published in [17]). TiO_2_ NWs were extremely long (greater than 10 μm), and the length could not be reliably measured.

Dynamic light scattering analysis conducted on 50 μg/mL NP suspensions in a cell culture medium consisting of serum shows that ZnO and CuO NPs have similar aggregate sizes of 323 and 337 nm, respectively. All TiO_2_ NPs showed values from 400–730 nm, except the TiO_2_ NIST 1898 NPs which aggregated to a larger extent with an average size of ~1250 nm. All suspensions were heterodisperse in the cell culture medium, with PDI values ranging from 0.39 for TiO_2_ NPs to 0.56 for ZnO NPs (Table 3).

### 3.2. Dissolution of ZnO and CuO Particles

With respect to ZnO and CuO particles, both are known to be soluble in biological environments. The dissolution of ZnO NPs and MPs was measured in the FE1 cell culture medium over 0–48 h (Figure 1). From the data presented, it can be seen that at the concentration of 100 µg/mL, both the ZnO NPs and MPs undergo immediate dissolution at a similar level. Concentration-dependant dissolution is seen with respect to the ZnO NPs, with the lower concentration showing a higher propensity to dissolve over the experimental time span (94.5% vs. 19.3% dissolved at 48 h for 10 and 100 µg/mL). The amount of dissolved material stays relatively consistent from 0–48 h for all Zn materials, albeit a slight increase over time can be seen for the 100 µg/mL concentration of ZnO NPs (14.6–19.3% dissolved from 0–48 h).

The CuO dissolution data were reproduced from [17] (Figure 1). From the data presented, it can be seen that the CuO particles exhibit dose, time, and size dependant dissolution from 0–48 h and dissolve to a greater extent than the same concentration of MPs (51.5% vs. 1.51% for NPs and MPs, respectively at 48 h). However, unlike ZnO, CuO NPs exhibit a higher degree of dissolution at higher concentrations (12.6% vs. 51.5% dissolved at 48 h for 10 and 100 µg/mL, respectively).

### 3.3. Viability Analysis of Cells Treated with NPs, MPs, and Metal Chlorides:

Trypan Blue exclusion assay was conducted to assess viability after 2–4 h MONM exposure. Phase-contrast images and percentage of cell viability results are shown in Figure 2, and Appendix A Appendix A. With respect to the ZnO and CuO, only the highest dose of ZnO NPs resulted in a statistically significant decrease in percentage viability as compared to control, down to 88% at 40 μg/mL (*p* < 0.05) after 4 h of exposure (Figure 2). From all the TiO_2_ forms assessed, only TiO_2_ 5424HT exposure resulted in a statistically significant loss of cell viability at 100 μg/mL at 2 h (89%, *p* < 0.05). Although at other doses of ZnO, there was no decrease in cell viability, phase-contrast images of ZnO NP, ZnO MP, and ZnCl_2_ showed significant cell rounding (Appendix A Appendix A) at both time points. Exposure to 100 μM H_2_O_2_ for 4 h did not result in a reduction of viability as compared to time-matched media controls (data not shown). Exposure to particle concentrations over 100 μg/mL resulted in significant particle overload for TiO_2_ NPs, which impaired accurate cell counting (data not shown). As such, doses over 100 μg/mL were not used for subsequent experiments.

### 3.4. Metal Oxide and Metal Chloride Induced DNA Damage

DNA damage, as assessed by the CometChip assay via percentage DNA in the tail, is most prominently seen with respect to soluble MONM exposures (Figure 3). Treatment with ZnO NPs induced dose- and time-dependent DNA damage, with 18 and 57% DNA in the tail at 2 and 4 h of exposure at 40 μg/mL (Figure 3A). The ZnO MPs induced significant DNA damage at all doses tested at both time points, with 12 and 52% DNA in the tail at 2 and 4 h at the dose of 40 μg/mL (Figure 3A). With respect to ZnCl_2_, dose- and time-dependent increases in DNA damage were seen but with a lower magnitude than that seen for ZnO NPs and ZnO MPs with equimolar amounts of metal. A maximal response of 7 and 27% DNA in the tail was observed in the 67 μg/mL ZnCl_2_ dose group at 2 and 4 h, respectively (Figure 3A). Similarly, at both post-exposure timepoints, CuO NPs induced dose- and time-dependent DNA strand breaks, with 24 and 58% DNA in the tail at 2 and 4 h, respectively, at 50 μg/mL. CuO MPs did not induce DNA damage at any dose or time point tested (Figure 3B). Treatment with CuCl_2_ did not induce any DNA damage (Figure 3B).

With respect to the insoluble TiO_2_ MONMs and MPs, the overall response was subtle compared to the responses observed in cells treated with ZnO and CuO NPs (Figure 3). TiO_2_ 5422 HT, 5423 HT, and R050P NPs exhibited a statistically significant increase in DNA damage at 50 and 100 μg/mL after 2 and 4 h of exposure (~10% DNA in tail). TiO_2_ 5422 HT induced a small but statistically significant response starting from 25 μg/mL at 2 h. The NIST TiO_2_ NPs were the least responsive, with ~5% DNA in the tail at 4 h, at doses of 50 and 100 μg/mL.

Media-only exposed negative controls consistently exhibited ~4.5% DNA in the tail and cells treated for 1 h with 100 μM H_2_O_2_ exhibited variation, with between 53–83% DNA in the tail.

### 3.5. DNA Damage Points of Departure for Metal Oxide and Metal Chloride Treated Cells

For the purposes of BMC modelling of comet data, the dose was expressed as either µg/mL of the material used (Figure 4) or as µg/mL of the constituent metal (Figure 5). The PODs (µg/mL material) determined through both BMC modelling with a BMR of 1.0 (100% increase in response over baseline), and classical methods are summarised in Table 4. Using univariate BMC modelling, it can be seen that there is distinct separation in potency at 2 h between CuO NPs, ZnO NPs, ZnO MPs, and ZnCl_2_, TiO_2_ 5423HT, TiO_2_ R050P with the following order of potency: CuO NP > ZnO NP > ZnO MP > ZnCl_2_, TiO_2_ 5423HT, TiO_2_ R050P (Figure 4). Covariate analysis at 2 h post-exposure, using the exposure compound as the covariate, increased confidence intervals for 2 samples and differences in BMC ranges became less distinguishable, although CuO NPs, ZnO NPs, and ZnO MPs consistently appeared more potent than ZnCl_2_ and the TiO_2_ NPs that showed dose–response. At 4 h, the trend in potency was largely the same as what was observed at the 2 h time point with respect to univariate modelling, albeit ZnCl_2_ appears distinctly more potent than the TiO_2_ NPs showing dose–response, exhibiting the following potency ranking: CuO NPs > ZnO NPs > ZnO MPs > ZnCl_2_ > TiO_2_ R050P, TiO_2_ 5423HT, TiO_2_ 5422HT. For the covariate analysis at the 4 h time point, Zn forms and Ti forms were treated as covariates. From the covariate analysis, there was no difference in potency between the three TiO_2_ NPs showing dose–response, while the ZnO NPs and MPs appeared more potent than ZnCl_2_ (Figure 4) In all BMC modelling instances, the NOEC/LOEC values were within the range of the lower and upper 95 percentile of the benchmark concentration (BMCL/BMCU) (Figure 4).

An additional dose metric was employed for BMC modelling, µg/mL of constituent metal, in order to determine whether the potency differences between the materials hold true when normalized to metal content (Figure 5). From the BMC modelling shown, it can be seen that the potency rankings largely remain the same, albeit a smaller separation in BMC ranges between the materials. At the 2 h timepoint, using univariate modelling the following potency ranking is obtained: CuO NP > ZnO NPs > ZnO MPs > TiO_2_ 5423HT, TiO_2_ R050P. The BMC interval for ZnCl_2_ overlapped with ZnO MPs, TiO_2_ 5423HT, and TiO_2_ R050P. Using covariate modelling, with all exposures as covariate, the separation in BMC intervals was lesser, with potency ranking showing CuO NPs, ZnO NPs, ZnO MPs > ZnCl_2_, TiO_2_ 5423HT, and TiO_2_ R050P. ZnCl_2_ also appeared more potent than TiO_2_ R050P. At the 4 h time point, the potency rankings for the univariate analysis are almost identical to results shown in Figure 4, with CuO NPs > ZnO NPs > ZnO MPs ~ ZnCl_2_ > TiO_2_ R050P, TiO_2_ 5423HT, TiO_2_ 5422HT, while covariate BMC modelling indicates ZnO NPs, ZnO MPs > ZnCl_2_ > TiO_2_ R050P, TiO_2_ 5423HT, TiO_2_ 5422HT. In all BMC modelling cases, the NOEC/LOEC values were within the BMCL/BMCU range (Figure 5).

## 4. Discussion

The genotoxic potential of the MONMs investigated in this study has been assessed previously, both in vitro and in vivo reviewed in [14,15,16]. While material size and solubility have been suggested to significantly impact the observed genotoxicity, systematic studies investigating the relative contributions of size and solubility to the levels of DNA damage to derive genotoxicity potency have not been conducted. In this study, the high-throughput CometChip assay was tested for its applicability for routine potency screening of NM induced genotoxicity using an in vitro lung cell culture model. Three MONMs were investigated including, ZnO, CuO, and TiO_2_ MONMs and MPs, as well as the dissolved metal equivalents ZnCl_2_ and CuCl_2_.

### 4.1. Genotoxicity of ZnO, CuO, and TiO_2_ MONMs

Genotoxicity resulting from ZnO, CuO, and TiO_2_ particle exposure has been consistently documented in a number of in vitro and in vivo model systems [14,15,16,38,39,40]. In A549 lung cells, pristine ZnO particles with a primary particle size of ~140 nm and aggregate sizes of 210 nm induced dose-dependent increases in percentage DNA in the tail in the comet assay, with statistically significant responses from 10 μg/cm^2^ after 3 h of exposure (~30 μg/mL concentration used in this study) [41]. In the same study, the authors also examined TiO_2_ NPs with an average size of ~110 nm with an aggregate size of ~220 nm in media and reported no statistically significant increase in DNA damage, although an increasing, non-significant, dose-dependent trend was observed [40]. More recently, the genotoxic potential of uncoated ZnO NPs (particle sizes ~ 16 nm) alongside uncoated TiO_2_ NPs (particles sizes < 25 nm) was investigated in human T-lymphocytes after 3–24 h exposure, and showed significant genotoxicity in the case of ZnO NPs (doses up to 100 µg/mL, both 3 and 24 h timepoints) but not TiO_2_ NPs [39]. With respect to the CuO NPs, using the same CuO NP type used in the present study (SA544868), Siivola et al., (2020) showed statistically significant increases in percentage DNA damage starting at a dose of 10 μg/cm^2^ (equivalent to 32 μg/mL in this study) after 3–24 h in BEAS-2B cells using the traditional alkaline comet assay [17,42]. The CuO MPs were shown to be genotoxic at the very high concentration of 50 μg/cm^2^. With respect to TiO_2_ NPs, literature is inconsistent with both negative and positive genotoxicity results reported for seemingly similar particles, although studies published between 2013–2020 note more positive responses than negative [14,20]. In the present study, CuO NPs were considerably more genotoxic compared to ZnO NPs but ZnO MPs induced DNA damage and not CuO MPs (Figure 3A,B). While TiO_2_ materials were largely inert, subtle but dose-dependent increases in DNA breaks were observed for some TiO_2_ NPs investigated (Figure 3C). The overall response profile of the ZnO and CuO NPs, and TiO_2_ materials generated in the high-throughput CometChip assay are in concordance with the reported literature.

For the soluble NPs—ZnO and CuO—cellular response depends upon both the dissolved and particulate fractions, which change in abundance over time. For ZnO NPs, dissolution experiments from the literature [18] as well as from this study (Figure 1) have shown that particles can undergo instantaneous dissolution in DMEM depending on the particle characteristics, with similar levels of dissolved Zn present in the suspension media at 0, 24, and 48 h post-suspension. On the other hand, CuO NPs undergo slower dissolution over time in DMEM, with ~3% dissolved at 0 h and up to 51% dissolved by 48 h post-suspension [17,36]. Comparatively, both CuO and ZnO MPs are shown to dissolve less in DMEM at the same post-suspension timepoints (Figure 1), which may be related to the decreased surface area for interaction as compared to their NP equivalents. The genotoxicity results in Figure 4 show that both CuO and ZnO NPs induce potent time and dose-dependent genotoxicity at 2 and 4 Hr, however for the MP and metal chloride exposures, only ZnO MPs and ZnCl_2_ initiated the DNA damage, and with lower potency than their respective NP (Table 4, Figure 4 and Figure 5). A study examining the genotoxicity of ZnO NPs, MPs, and ZnCl_2_ in MDCK kidney cells following 24 h exposure using the comet assay only reported positive results with respect to the NPs [43]. Similar responses were seen in A549 cells exposed to ZnO NPs and ZnCl_2_ for 24 Hr, where only the ZnO NPs induced significant induction of double-strand breaks [44]. Both studies indicate that toxicity is induced by Zn uptake, however, the magnitude of response cannot be explained by the dissolved fraction alone. In another study examining the genotoxic potential of CuO NPs, CuO MPs, and CuCl_2_ via alkaline DNA unwinding, only the CuO NPs were able to induce DNA damage after 24 h of incubation [45]. This was proposed to be due to high levels of Cu detected in the cell nucleus after NP exposure, as compared to MPs and CuCl_2_. This was also observed in another study focusing on these three CuO species, with CuO NPs producing significantly higher Cu levels in the nucleus and cytoplasm than CuCl_2_ and CuO MPs after 24 h of exposure [46]. Together, in alignment with published in vitro genotoxicity literature, high-throughput CometChip assay results shown in Figure 4 and Figure 5 show evidence that the NP fractions are crucial to the genotoxic response in FE1 cells, but suggest that for ZnO NPs, the ionic fraction is a critical contributor to the observed DNA damage compared to dissolved Cu from CuO NP. Although not investigated in this study, the results suggest that Cu ions are slower to be internalised compared to Zn ions and may involve different membrane transport mechanisms, impacting their uptake and intracellular availability for reactions.

Unlike soluble ZnO and CuO NPs, TiO_2_ MONMs are insoluble and induce genotoxicity through interactions with the particle surface and through the formation of ROS or reactive nitrogen species [14,21]. Cellular responses, in this case, are dependent on particle properties such as size, crystallinity, and surface coating/functionalization, although a thorough assessment of the impact of surface coating on TiO_2_ genotoxicity is lacking [21]. In this study, the genotoxicity of nine different TiO_2_ MONMs and a TiO_2_ MP were examined, five of which are surface coated and five of which are pristine (Table 1 and Table 2). For the coated particles, four are coated in silica or silica in conjunction with another compound, and one is coated in a manufacturer-described ‘hydrophilic’ coating. With respect to particle size, of the 10 materials tested, 5 are ~50 nm in size, 2 are ~30 nm, 1 is ~5 nm, 1 is ~1500 nm, and another is a high-aspect ratio NW. Finally, 10 TiO_2_ NMs comprised of 3 different crystalline configurations—6 consisting of anatase/rutile mixture, 2 anatase, and 1 rutile only in composition (Table 1 and Table 2). With respect to the CometChip screening of the 10 different TiO_2_ metal oxides, only 3 induced a dose-dependant increase in DNA damage at either time point of assessment (Table 4). For two variants (5422HT, 5423HT), the surface was coated with silica or a silica–alumina formulation and both particle types had measured primary sizes of ~30 nm, with a predominant anatase crystal phase. The third type (R050P) that induced a dose-dependent DNA damage response is an uncoated type, with a primary particle size of ~50 nm and a rutile crystal phase. In all three cases, the response did not exceed 10% DNA in the tail indicating a relatively subtle potential to induce DNA damage among the active TiO_2_ forms assessed. Research into the toxicity of silica has shown that surface silanol groups and siloxane bridges are key mechanistic players involved in its pathogenicity, with the reactivity of these surface groups impacting endpoints such as DNA damage, and cytotoxicity through membranolysis [47], providing a potential explanation for the activity of TiO_2_ 5422HT and 5423HT. With respect to crystallinity, the rutile crystal arrangement has exhibited greater toxic potential than anatase both in vitro and in vivo [48,49], although contradictory results do exist [20]. Analysis of radicals generated from TiO_2_ particles indicates differential reactivity based on crystal structure, with rutile being able to catalyze the formation of hydroxyl radicals under light occlusion, while anatase induces the formation of superoxide anions [50]. This may potentially explain the significant (but small) dose-dependent response of TiO_2_ R050P, as this particle was uncoated and entirely of rutile configuration (Table 4). Currently, there is no consensus in the literature concerning the impact of coating, particle size, or crystallinity on the genotoxicity of TiO_2_ (recent positive and negative studies summarised in [14,20,21]). However, results from the high-throughput CometChip screening indicate a rutile configuration and silica coating increase the material’s DNA damage potential. While the results of the present study reflect acute responses, long-term studies using advanced cell culture models may be required to truly evaluate the potential of 5422HT, 5423HT and R050P.

In addition to mechanistic research using in vitro systems, all three MONMs have been used to assess pulmonary effects in vivo [49,51,52,53,54,55]. In Wistar rats, inhalation of CuO NPs and ZnO NPs has been shown to result in an acute inflammatory response, punctuated by inflammation and damage of alveolar and bronchiolar tissue [54,55]. The effects of these are thought to be mediated by the intracellular dissolution of the particle within acidic lysosomes [51,54]. With respect to ZnO exposure, a recovery period has been shown to reduce the inflammatory response, however, detectable levels of inflammatory markers are still present 15 days post-exposure [55], and 4 weeks post-exposure [51] depending on the exposure conditions. Using a dose range relevant for occupational exposures, inflammation and lung damage resulting from CuO NP exposure in rats was shown to largely ameliorate by 22 days post-exposure, with the exception of the highest dose used [54]. With respect to TiO_2_ NPs, intratracheal instillation in mice has been shown to result in inflammatory and fibrotic pathologies, with crystal phase, particle size, shape, surface area, and surface composition shown to be important predictors of response [49,53]. Of note, even though inflammation and lung damage was noted after intratracheal exposure to TiO_2_ NPs, no pronounced DNA damage as assessed by comet assay was noted in BALF cells, lung cells, or liver cells after 1, 3, 28, or 90 days post-exposure [53]. Thus, while some of these materials may be DNA damage-inducing, the long-term impact of early DNA damage remains to be assessed.

### 4.2. Mechanisms of Toxicity Underlying CuO, ZnO, and TiO_2_ Responses

With respect to CuO and ZnO, genotoxicity is thought to be induced through interactions of both the particulate and dissolved fractions through (1) the formation of ROS, (2) direct interaction with the DNA or DNA maintenance machinery by gaining access to the nuclear compartment, or (3) during mitosis where the nuclear membrane dissolves [56]. For these particles, toxicity is suggested to act via the ‘Trojan Horse’ mechanism [51,57,58], which involves active transport of the particulate ZnO and CuO into the cells (typically via endocytotic pathways), sequestered in acidic vesicles before ending up in the lysosomes and undergoing rapid dissolution. Intravesicular dissolution releases large amounts of metal ions within the lysosomal lumen, changing the pH of the microenvironment. The dissolved particles generate ROS and cause damage to the integrity of the vesicular membrane, resulting in vesicular rupture and release of the contents into the cytoplasm. Consequently, antioxidant defences are overwhelmed, resulting in damage to biomolecules (proteins, lipids, nucleic acids) either directly or through ROS imbalances. These effects culminate in genotoxicity and cytotoxicity if left unchecked. With respect to ZnO, dissolution results in Figure 1 seem to indicate that the Trojan Horse mechanism may have less of an influence on toxicity than previously surmised and that the dissolved fraction is the main mediator of genotoxicity (Figure 3A). It is important to note that ZnO NPs are less genotoxic but more cytotoxic compared to CuO NPs, suggesting the underlying mode of action for the two responses may be different. In one of our recent publications involving the same CuO NPs, CuO MPs, and CuCl_2_ compound used in this study, a differential impact of size and ionic fraction on transcriptional response was shown [17]. Both CuO NPs and CuCl_2_ were shown to induce both oxidative stress responses and DNA damage responses at the canonical pathway level, however, the CuCl_2_ mediated response was shown to have a slower onset than the CuO NP mediated response, and induced cytotoxicity to a lesser extent. Although no genotoxicity was reported for CuCl_2_ for up to 4 h as assessed by the CometChip assay in the present study, the possibility that DNA damage may be seen at a longer exposure duration cannot be disregarded.

TiO_2_ is insoluble and interplay between dissolved and particulate fractions is not anticipated, precluding a ‘Trojan Horse’ mechanism of action or metal-ion-induced toxicity. Instead, genotoxicity induced by TiO_2_ can proceed through the formation of ROS from the particle surface, interaction with the DNA in the nucleus, and also through interaction with mitosis/DNA maintenance and repair machinery [56]. With respect to the formation of ROS, the crystal configuration (anatase vs. rutile) dictates the reactivity of the particle, with anatase showing enhanced reactivity under UV irradiation [59] although both crystal configurations can generate radicals even under light occlusion [50]. These reactive molecules can induce oxidative DNA damage in an indirect manner. In addition to ROS formation, TiO_2_ NPs within the cell have been shown to interact with the mitotic spindle in vitro, resulting in improper chromosomal segregation and the formation of micronuclei [60]. It has also been proposed that TiO_2_ particle loaded vesicles can impinge upon and deform the nucleus, resulting in chromosomal damage and the formation of micronuclei [61]. Results from the DNA damage screening (Figure 3C) do not provide indications for any of the above mechanisms, and while TiO_2_ aggregates can be seen associated with exposed FE1 cells (Appendix A Appendix A), their subcellular localization cannot be determined. Due to the particle size of the three DNA damage-inducing TiO_2_ NPs (30–50 nm, Table 1), direct nuclear access is unlikely. Therefore, TiO_2_ NP DNA damage noted in this study may have proceeded through ROS production, interaction with DNA during mitosis when the nuclear membrane is dissolved, or through interaction with mitosis/DNA repair and maintenance machinery. It is important to note that while responses noted are minimal, it is possible that additional DNA damage may result from longer exposure regimens.

### 4.3. Benchmark Concentration Modelling and Relative Potency Ranking of MONMs

Classical methods for determining points of departure (PODs) include the lowest-observed-effect-concentration (LOEC) and the no-observed-effect-concentration (NOEC), which are based upon point estimates of endpoint response and do not rely on the whole concentration-response relationship to determine the POD. In contrast, BMC modelling makes use of the entire concentration-response relationship and determines a POD (known as the BMC) with confidence intervals related to the variation inherent in the dataset used to conduct modelling. Within PROAST, it is possible to conduct BMC modelling with or without using a covariate approach. If the underlying data structure is adequate, covariate analysis can produce tighter BMC confidence intervals, as highlighted in [62].

Currently, the preferred method of deriving PODs is the benchmark modelling approach [63], which has recently been used to rank 28 chemicals for their potential to induce in vitro DNA damage in the CometChip assay [64]. A similar approach was utilized here to conduct a potency ranking of DNA damage induced by dose-responding metal oxides and chlorides at 2 and 4 h. A BMR of 1.0, with or without covariate analysis was used, with mass-concentrations of the compound (Figure 4) or constituent metal (Figure 5) as dose metrics. Covariate analysis has been suggested to increase the precision of BMC estimation, as long as response data at each level of the covariate can be described using models with constant shape parameters [62]. A noticeable increase in precision (BMCL–BMCU range) was seen between univariate and covariate analyses with respect to dose-responding TiO_2_ at 4 h, although this effect was less pronounced, and even opposite, with respect to the Zn forms and CuO NPs at both time points (Table 4, Figure 4 and Figure 5). The covariate approach was not uniformly applicable across exposures due to inherent differences in dose–response structures, and further subdivision of exposures at the 4 h time point was necessary (Figure 4 and Figure 5; TiO_2_ nanoforms and Zn forms as covariates). Due to the difficulties in uniformly applying the covariate approach, potency ranking and trends seen are based on univariate modelling only.

With respect to compound-based potency ranking, a similar rank amongst the dose-responding exposures is seen at both time points; although the differences in potency were more distinct at 4 h, with the following trend seen: CuO NPs (SA544868) > ZnO NPs (US3580) > ZnO MPs (US1003M) > ZnCl_2_ > TiO_2_ R050P, TiO_2_ 5423HT, TiO_2_ 5422HT. Expressing the dose as a function of the constituent metal concentration markedly decreases the difference in potency between the various compounds (Figure 5), although a similar trend is retained at the 4 h time point: CuO NPs (SA544868) > ZnO NPs (US3580) > ZnO MPs (US1003M) ~ ZnCl_2_ > TiO_2_ R050P, TiO_2_ 5423HT, TiO_2_ 5422HT. The ranking trends indicate that 1) soluble MONMs induce more pronounced DNA damage than insoluble MONMs, 2) dissolved ions contribute substantially to the observed damage of ZnO, and 3) the DNA damage induced by TiO_2_ nanoforms is subtle and is not specific to material properties. This study represents one of the first to use the CometChip assay to show differential potency of soluble and insoluble particles to induce DNA damage. Some results are in alignment with published literature [65,66]; however, the results from this study suggest that soluble NPs differ in the toxicity mode of action and in their relative potency, implying further studies are needed before the application of read-across strategies in risk assessment of soluble MONMs.

As compared to the traditional alkaline comet assay, the CometChip assay allows for greatly enhanced throughput with a reduced labour cost, which makes it an attractive genotoxicity test for NM. The increase in throughput allowed for 20 separate particle exposures to be conducted on one single plate, with four technical replicates per condition, which produced enough information to allow for BMC modelling and relative potency estimation. Thus, this assay can be used in tier-1 testing strategies and screening approaches to identify potentially hazardous NM. In all, this study provides support for the use of the CometChip assay for routine in vitro screening of NM genotoxic potential.

## 5. Conclusions

In conclusion, this study provides evidence for the applicability of the commercially available CometChip assay for routine screening of NM induced DNA damage. The ZnO, CuO, and select TiO_2_ NPs investigated exhibited dose- and time-dependent DNA damage, with maximal responses of 58, 57, and 10% DNA in the tail, respectively, at 4 h. The results of dose–response modelling highlight the differential impact of the dissolved and particulate fractions on the DNA damage potential of CuO and ZnO, as well as the importance of surface coating and crystallinity on genotoxicity induced by TiO_2_. With the increasing number and variety of NM and MONMs in use, traditional low-throughput genotoxicity testing methods are insufficient to meet regulatory needs for their assessment. By leveraging mechanistically based high-throughput in vitro screening assays, a large number of relevant endpoint-specific data can be generated which can aid in prioritization, grouping, and read-across endeavours.

## Figures and Tables

**Figure 1 nanomaterials-12-01844-f001:**
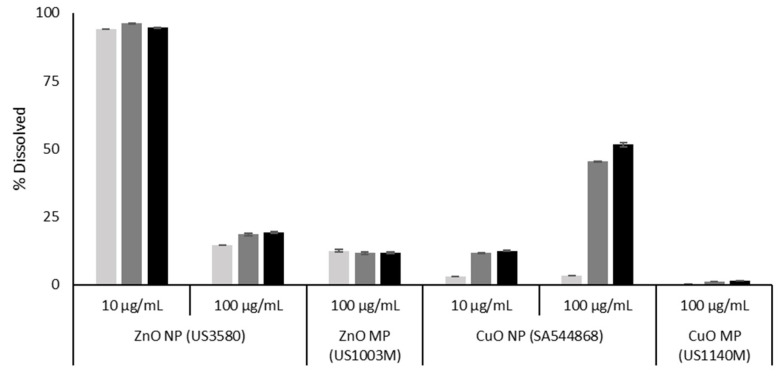
Percentage dissolution of CuO and ZnO NPs (US3580; SA544868) and MPs (US1003M; US1140M) in DMEM F12 cell culture media, with 2% serum, after 0–48 h of incubation at 37 °C. Error bars represent ± standard deviation (*n* = 3). Light grey: 0 h. Dark grey: 24 h. Black: 48 h. CuO data were reproduced from [17].

**Figure 2 nanomaterials-12-01844-f002:**
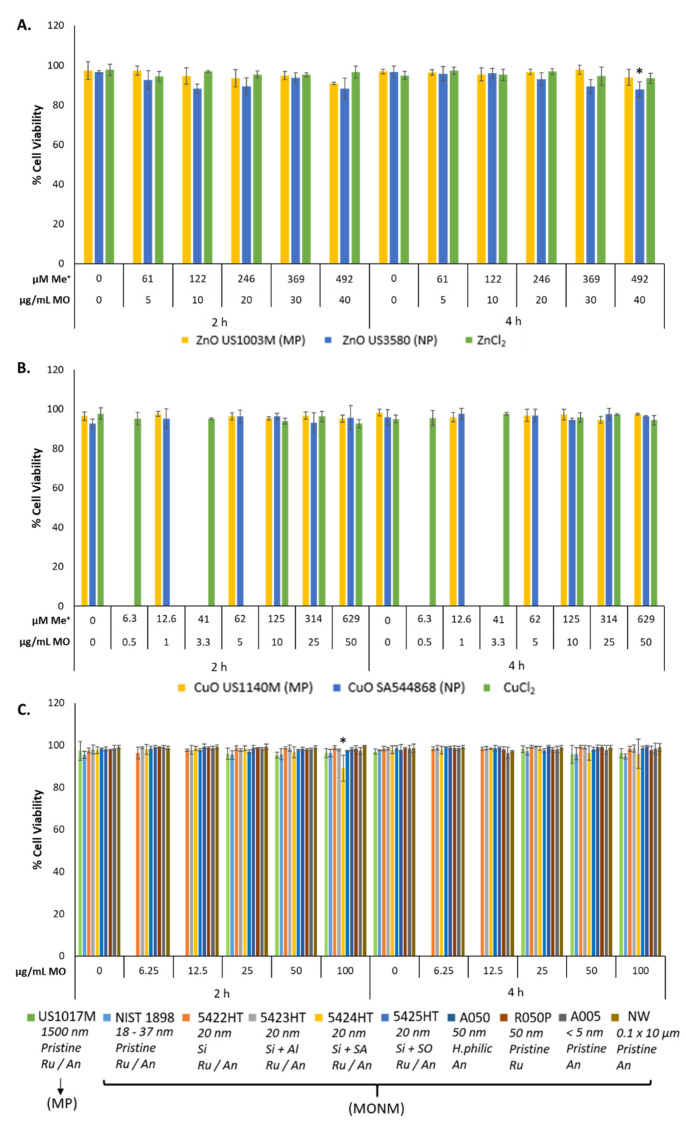
FE1 percentage cell viability analysis following 2–4 h exposure to (**A**) Zn forms, (**B**) Cu forms, and (**C**) TiO_2_ NPs and MPs. Statistically significant differences between the exposed samples and the media control were determined through a one-way ANOVA with a Dunnett’s post hoc in the case of significant results. * *p* < 0.05. Error bars are +/−SD (*n* = 3–4). µM Me+: dose expressed in terms of micromolarity of the constituent metal. µg/mL MO: dose expressed as mass concentration of metal oxide. NW: nanowire, Ru/An: anatase + rutile mix, An: anatase, Ru: rutile, Si: silica coated, Si + Al: silica and alumina coated, Si + SA: silica and stearic acid coated, Si + SO; silica + silicone oil, H.philic: hydrophilic coating.

**Figure 3 nanomaterials-12-01844-f003:**
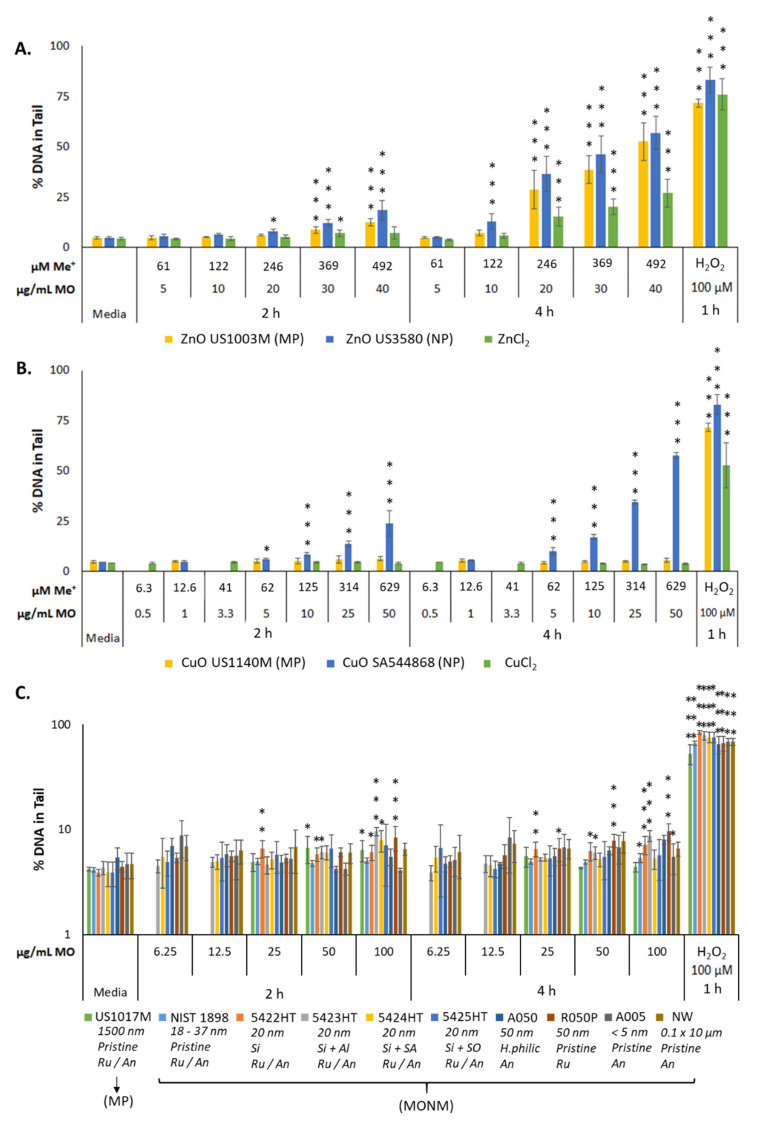
DNA damage, as measured by percentage DNA in tail, in FE1 cells after 2–4 h exposure to (**A**) Zn forms, (**B**) Cu forms, and (**C**) TiO_2_ materials. Statistically significant differences between the exposed samples and the media control were determined through a one-way ANOVA with a Dunnett’s post hoc in the case of significant results. * *p* < 0.05. ** *p* < 0.01. *** *p* < 0.001. Error bars are ±SD (*n* = 3–4). NW: nanowire, Ru/An: anatase + rutile mix, An: anatase, Ru: rutile, Si: silica coated, Si + Al: silica and alumina coated, Si + SA: silica and stearic acid coated, Si + SO; silica + silicone oil, H.philic: hydrophilic coating.

**Figure 4 nanomaterials-12-01844-f004:**
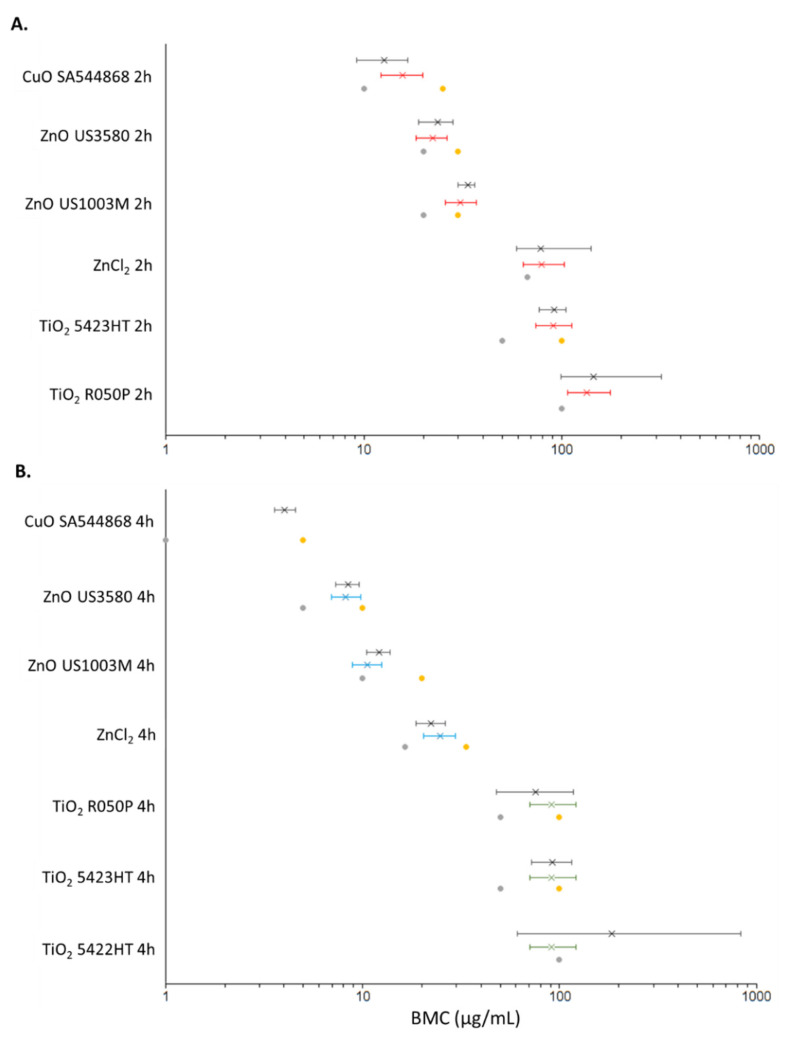
BMC plots showing the results of PROAST BMC modelling (BMR = 1.0, 100% increase over baseline) at 2 h (**A**) and 4 h (**B**) for metal oxides and metal chlorides which exhibit a dose–response. Left error bars: BMCL. Right error bars: BMCU. X: BMC. Grey dot: NOEC. Yellow dot: LOEC. Red: BMC modelling with exposure type (NP, MP, metal chloride) as covariate. Black: BMC modelling with no covariate. Blue: BMC modelling with Zn type as covariate. Green: BMC modelling with TiO_2_ type as covariate.

**Figure 5 nanomaterials-12-01844-f005:**
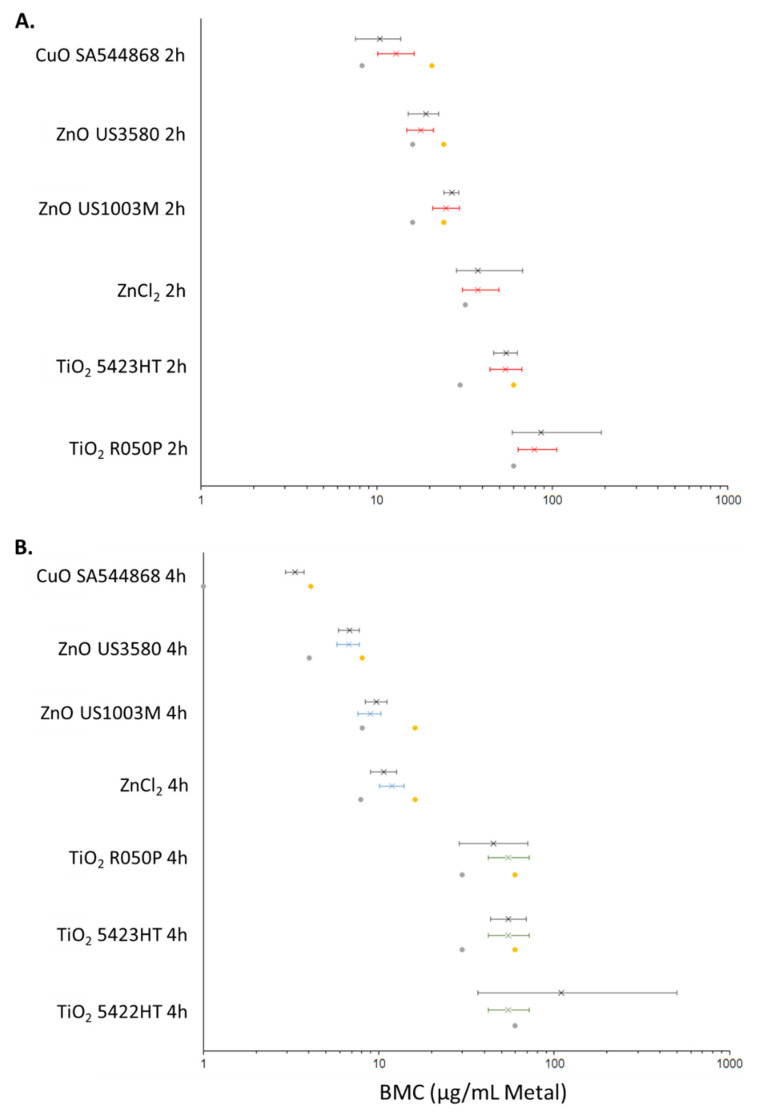
BMC plots showing the results of PROAST BMC modelling (BMR = 1.0, 100% increase over baseline) at 2 (**A**) and 4 (**B**) h for metal oxides and metal chlorides which exhibit a dose–response. The dose is represented in terms of µg/mL of the constituent metal. Left error bars: BMCL, lower 95 percentile or BMC. Right error bars: BMCU, upper 95 percentile of BMC. X: BMC. Grey dot: NOEC. Yellow dot: LOEC. Red: BMC modelling with exposure type (NP, MP, metal chloride) as covariate. Black: BMC modelling with no covariate. Blue: BMC modelling with Zn type as covariate. Green: BMC modelling with TiO_2_ type as covariate.

**Table 1 nanomaterials-12-01844-t001:** Particle characterization of the CuO, ZnO, and TiO_2_ NMs used for genotoxicity screening. Measured primary NP sizes are presented in terms of width and length, with standard deviation in parentheses for each respective measurement (*n* = 100–200). For TiO_2_ NW (774510) only the width was directly measured due to the tangled nature of the fibres. PPS: primary particle size. SSA: specific surface area.

Metal Oxide	Manufacturer (Catalogue Number)	Coating	PPS Reported ^a^	PPS Measured (Standard Deviation)	Aspect ratio (Standard Deviation)	SSA (m^2^/g) ^a^
ZnO	US Research Nanomaterials Inc. (US3580)	Pristine	35-45 nm	23.9 × 19.4 nm (7.2 × 5.5 nm)	1.23 (0.17)	65
CuO	Sigma Aldrich (544868)	Pristine	28 nm	64.8 × 45.9 nm ^b^ (28.0 × 47.0 nm)	1.39 (0.39)	33
TiO_2_	National Institute of Standards and Technology (1898)	Pristine	19 nm Anatase (76%) 37 nm Rutile (24%)	26.8 × 20.8 nm (8.9 × 6.8 nm)	1.30 (0.26)	55.55
TiO_2_	Nanostructured & Amorphous Materials, f. (5422HT)	Silica	20 nm (80–90% Anatase; 10–20% Rutile)	52.6 × 32.7 nm (25.3 × 12.4 nm)	1.63 (0.51)	≥40
TiO_2_	Nanostructured & Amorphous Materials, Inc. (5423HT)	Silica & Alumina	20 nm (80–90% Anatase; 10–20% Rutile)	37.6 × 30.1 nm (23.1 × 18.3 nm)	1.32 (0.71)	≥40
TiO_2_	Nanostructured & Amorphous Materials, Inc. (5424HT)	Silica & Stearic Acid	20 nm (80–90% Anatase; 10–20% Rutile)	49.5 × 32.9 nm (20.4 × 12.4 nm)	1.54 (0.42)	≥10
TiO_2_	Nanostructured & Amorphous Materials, Inc. (5425HT)	Silica & Silicone Oil	20 nm (80–90% Anatase; 10–20% Rutile)	50.3 × 32.5 nm (42.8 × 13.3 nm)	1.53 (0.75)	≥10
TiO_2_	MKNano (MK-TiO_2_-A050)	Hydrophilic	50 nm (Anatase)	30.7 × 26.3 nm (8.57 × 7.01 nm)	1.17 (0.15)	N/A
TiO_2_	MKNano (MKN-TiO_2_-R050P)	Pristine	50 nm (Rutile)	82.4 × 52.9 nm (34.6 × 14.6 nm)	1.56 (0.51)	N/A
TiO_2_	MKNano (MKN-TiO_2_-A005)	Pristine	<5 nm (Anatase)	7.6 × 5.4 nm (2.1 × 1.6 nm)	1.42 (0.30)	356
TiO_2_	Sigma Aldrich (774510)	Pristine	100 nm × 10 000 nm	106.5 nm (147.5 nm)	----	N/A

^a^ Information obtained from manufacturer. ^b^ Information obtained from [17].

**Table 2 nanomaterials-12-01844-t002:** Particle characterization information of the ZnO, CuO, and TiO_2_ MPs used for genotoxicity screening. PPS: primary particle size. SSA: specific surface area.

Metal Oxide	Manufacturer (Catalogue Number)	Coating	PPS Reported (nm) ^a^	SSA (m^2^/g) ^a^
ZnO	US Research Nanomaterials Inc. (US1003M)	Pristine	1000	2−5.8
CuO	US Research Nanomaterials Inc. (US1140M)	Pristine	5000	4−6
TiO_2_	US Research Nanomaterials Inc. (US1017M)	Pristine	1500 nm Anatase 1500 nm Rutile	5−8

^a^ Information obtained from manufacturer.

**Table 3 nanomaterials-12-01844-t003:** Dynamic light scattering analysis of MONPs within DMEM cell culture media +2% fetal bovine serum.

Metal Oxide	Manufacturer (Catalogue Number)	Hydrodynamic Diameter (50 µg/mL DMEM)	Polydispersity Index (50 µg/mL DMEM)
ZnO	US Research Nanomaterials Inc. (US3580)	323 ± 125 nm	0.56 ± 0.12
CuO	Sigma Aldrich (544,868)	337 ± 17.4 nm	0.40 ± 0.04
TiO_2_	National Institute of Standards and Technology (1898)	1334 ± 48.0 nm	0.22 ± 0.06
TiO_2_	Nanostructured & Amorphous Materials, Inc. (5422HT)	663 ± 49.7 nm	0.35 ± 0.05
TiO_2_	Nanostructured & Amorphous Materials, Inc. (5423HT)	726 ± 85.3 nm	0.31 ± 0.06
TiO_2_	Nanostructured & Amorphous Materials, Inc. (5424HT)	563 ± 23.6 nm	0.41 ± 0.05
TiO_2_	Nanostructured & Amorphous Materials, Inc. (5425HT)	553 ± 27.2 nm	0.33 ± 0.04
TiO_2_	MKNano (MK-TiO_2_-A050)	460 ± 66.6 nm ^a^	0.37 ± 0.07 ^a^
TiO_2_	MKNano (MKN-TiO_2_-R050P)	373 ± 11.3 nm	0.27 ± 0.02
TiO_2_	MKNano (MKN-TiO_2_-A005)	407 ± 17.6 nm	0.23 ± 0.02

^a^ DMEM + 0.02 mg/mL BSA.

**Table 4 nanomaterials-12-01844-t004:** Classical and benchmark concentration modelling to derive points of departure (µg/mL) based upon percentage DNA in tail measurements after 2–4 h exposure to NPs, MPs, and metal chlorides. For all BMC modelling, the benchmark response was set to 1.0 (100% extra risk). NOEC: no-observed-effect-concentration. LOEC: lowest-observed-effect-concentration. BMC: benchmark concentration. BMCL: the lower 95% confidence interval of the BMC. BMCU: the upper 95% confidence interval of the BMC. No Covariate: All BMC analyses were computed for each individual particle and timepoint. Covariate: BMC modelling conducted using covariate approach.

Timepoint	Metal Oxide	Particle Type	Classical		BMC Modelling (No Covariate)			BMC Modelling (Covariate)		
			NOEC ^a^	LOEC ^b^	BMCL	BMC	BMCU	BMCL	BMC	BMCU
2 h	ZnO	US3580	20	30	18.9	23.7	28.2	18.4 ^c^	22.3 ^c^	26.3 ^c^
		US1003M	20	30	29.9	33.5	36.5	25.9 ^c^	30.9 ^c^	37 ^c^
		ZnCl_2_	67	>67	59.2	78.7	141	64.2 ^c^	79.1 ^c^	103 ^c^
	CuO	SA544868	10	25	9.16	12.7	16.7	12.2 ^c^	15.7 ^c^	19.8 ^c^
		US1140M	50	>50	-	-	-	-	-	-
		CuCl_2_	108	>108	-	-	-	-	-	-
	TiO_2_	NIST1898	100	>100	-	-	-	-	-	-
		5422HT	100	>100	-	-	-	-	-	-
		5423HT	50	100	77.1	91.4	105	73.9 ^c^	90.4 ^c^	112 ^c^
		5424HT	100	>100	-	-	-	-	-	-
		5425HT	100	>100	-	-	-	-	-	-
		A050	100	>100	-	-	-	-	-	-
		R050P	100	>100	99	145	318	107 ^c^	134 ^c^	176 ^c^
		A005	100	>100	-	-	-	-	-	-
		Nano Wires	100	>100	-	-	-	-	-	-
		US1017M	100	>100	-	-	-	-	-	-
4 h	ZnO	US3580	5	10	7.32	8.48	9.6	6.94 ^d^	8.24 ^d^	9.77 ^d^
		US1140M	10	20	10.5	12.1	13.8	8.87 ^d^	10.6 ^d^	12.5 ^d^
		ZnCl_2_	16.5	33.5	18.7	22.3	26.2	20.5 ^d^	24.9 ^d^	29.6 ^d^
	CuO	SA544868	1	5	3.56	4.04	4.55	-	-	-
		US1003M	50	>50	-	-	-	-	-	-
		CuCl_2_	108	>108	-	-	-	-	-	-
	TiO_2_	NIST1898	100	>100	-	-	-	-	-	-
		5422HT	100	>100	61.1	184	832	70.4 ^e^	91.5 ^e^	121 ^e^
		5423HT	50	100	72.3	91.8	115	70.4 ^e^	91.5 ^e^	121 ^e^
		5424HT	100	>100	-	-	-	-	-	-
		5425HT	100	>100	-	-	-	-	-	-
		A050	100	>100	-	-	-	-	-	-
		R050P	50	100	47.7	75.4	118	70.4 ^e^	91.5 ^e^	121 ^e^
		A005	100	>100	-	-	-	-	-	-
		Nano Wires	100	>100	-	-	-	-	-	-
		US1017M	100	>100	-	-	-	-	-	-

^a^ NOEC is defined as the first dose preceding the LOEC. ^b^ LOEC is defined as the first dose where there is a statistically significant (*p* < 0.05) increase in percentage DNA in the tail of at least 2-fold as compared to the media control. ^c^ Exposure type set as covariate. ^d^ Zn exposure type set as covariate. ^e^ TiO_2_ particle type set as covariate.

## Data Availability

The data presented in this study are available on request from the corresponding author.

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
