# Peer review of "The High-Throughput In Vitro CometChip Assay for the Analysis of Metal Oxide Nanomaterial Induced DNA Damage"

_nanomaterials, 2022, doi:10.3390/nano12111844_

Round 1
Reviewer 1 Report
Please see the atachment.

Author Response
Reviewer 1:
The paper “The high-throughput in vitro CometChip assay for the analysis of metal oxide nanomaterial induced DNA damage” describes the possibility to use the high-throughput CometChip assay as a screening tool for a wide range of nanomaterials (NM). The authors have applied the high-throughput CometChip assay and evidenced the role of size, solubility and surface coatings on NM induced genotoxicity. The DNA damage in adherent murine lung epithelial (FE1) cells exposed to different doses of ZnO, CuO, and TiO2 NPs, MPs as well as zinc and copper chloride salts were observed to be both dose and time dependant. No-observed-effect-concentration (NOEC) and the lowest-observed-effect concentration (LOEC) were connected with benchmark concentration (BMC) modelling to determine points of departure (PODs) in order to evidence the differences in relative potency between the various exposures. The particles were characterised by TEM and the ZnO dissolution was determined with a proper method.
Considering the risks associated with exposure to such materials it is important to develop a proper method to determine their effect on DNA damage for a proper protection of the persons that manipulate or use NM.
I therefore recommend minor revision having in view the following aspects:
- The ZnO and CuO NPs solubility in water is poorly so this must be connected either to
pH or to biological medium in Introduction part.
Response:
The solubility of ZnO and CuO NPs have been measured in biological medium in the past. The introduction section was slightly modified as follows to highlight this:
“Both ZnO and CuO NPs are soluble in biological medium [42, 17], and their toxicity is thought to be a combination of both particulate and dissolved metal species. The microparticle (MPs) counterparts of CuO and ZnO NP types are shown to be less soluble [17 – 18] and differ in their toxicity potential compared to their nano forms.”
- (Table or Figure x.) must be corrected as (Table or Figure x) in whole paper.
Response:
“Table or Figure x.” was amended to “Table or Figure x” throughout the manuscript.
- The sentence “The genotoxic potential of the MONMs investigated in this study has
been assessed previously, both in vitro and in vivo.” requests reference(s).
Response:
The sentence was modified as follows:
“The genotoxic potential of the MONM investigated in this study has been assessed previously, both in vitro and in vivo [reviewed in 14 – 16]”

Reviewer 2 Report
Proposed paper constitutes a well-prepared, interesting and well-presented work. All parts of the manuscript look adequately, i.e. from well-explained importance of undertaken research topic, well-described methodology and results of performed experiments to adequately reported conclusions. The only suggested minor revisions are given in a more detail below:
- Abstract of the paper should be supplemented with the novelty of the research and a brief background.
- Abstract of the paper should be supplemented with the novelty of the research and a brief background.
- Final Conclusions should be more quantified.
Author Response
Reviewer 2:
Proposed paper constitutes a well-prepared, interesting and well-presented work. All parts of the manuscript look adequately, i.e. from well-explained importance of undertaken research topic, well-described methodology and results of performed experiments to adequately reported conclusions. The only suggested minor revisions are given in a more detail below:
- Abstract of the paper should be supplemented with the novelty of the research and a brief background.
- Abstract of the paper should be supplemented with the novelty of the research and a brief background.
Response:
The abstract was modified to include a brief background, as well as the main novel aspect of the manuscript:
“Metal oxide nanomaterials (MONMs) are among one of the most highly utilized classes of nanomaterials worldwide, though their potential to induce DNA damage in living organisms is known. High-throughput in vitro assays have the potential to greatly expedite analysis and understanding of MONM induced toxicity, while minimizing the overall use of animals. In this study, the high-throughput CometChip assay was used to assess the in vitro genotoxic potential of pristine copper oxide (CuO), zinc oxide (ZnO), and titanium dioxide (TiO2) MONMs and microparticles (MPs) as well as 5 coated / surface modified TiO2 NPs and zinc (II) chloride (ZnCl2) & copper (II) chloride (CuCl2) after 2 – 4 hr of exposure. The CuO NPs, ZnO NPs & MPs, and ZnCl2 exposures induced dose- and time-dependent increases in DNA damage at both timepoints. TiO2 NPs surface coated with silica or silica-alumina and one pristine TiO2 NPs of rutile crystal structure also induced subtle dose-dependent DNA damage. Concentration modelling at both post-exposure timepoints, highlighted the contribution of the dissolved species to the response of ZnO, and the role of the nanoparticle fraction for CuO mediated genotoxicity, showing the differential impact that particle size, chemical composition and dissolved fractions can have on genotoxicity induced by MONMs. The results imply that solubility alone may be insufficient to explain the biological behavior of MONMs.
- Final Conclusions should be more quantified.
Response:
The conclusions are modified to incorporate a more quantitative summary of the responses along side the broad conclusions of the paper as a whole:
“In conclusion, this study provides evidence for the applicability of the commercially available CometChip assay for routine screening of NM-induced DNA damage. The ZnO, CuO, and select TiO2 NPs investigated exhibited dose and time dependant DNA damage, with maximal responses of 58, 57, and 10% DNA in tail, respectively, at 4 h. The results of dose-response modelling highlight the differential impact of the dissolved and particulate fractions on the DNA damage potential of CuO and ZnO, as well as the importance of surface coating and crystallinity on genotoxicity induced by TiO2. With the increasing number and variety of NM and MONMs in use, traditional low-throughput genotoxicity testing methods are insufficient to meet regulatory needs for their assessment. By leveraging mechanistically based high-throughput in vitro screening assays, large number of relevant endpoint-specific data can be generated, which can aid in prioritization, grouping, and read-across endeavours.

Reviewer 3 Report
The manuscript investigated the DNA damage toxicity of several metal oxide nanomaterials (ZnO, CuO, TiO2) in alveolar epithelial cells and evaluated their in vitro genotoxicity by high-throughput CometChip assay. The research is less studied and too much discussed, which makes the paper more of a review than a research paper. In addition, I have the following suggestions and questions:
- I wonder that does exposure to MONMs cause it to accumulate in the lungs and cause damage? To be more clear, does MONMS make direct contact with alveolar epithelial cells?
- 2. Each paragraph of results should have a corresponding result diagram produced by the author in the manuscript, not in the supplement or have a list from known information, such as 3.1 and 3.3.
- 3. It is recommended to segment the ordinate of Figure 2C to show the difference better.
- 4. It is more meaningful to caption the material property (such as crystal type and Particle size) than to show the Catalogue Number in figuer2.
- It is suggested to add a discussion about the dose of human exposure as to whether the dose of human exposure can be inferred from the dose of DNA damage limited to cells.
- Since it is a high-throughput in vitroCometChip assay, why not add the corresponding cometassay pictures in the manuscript?
Author Response
Reviewer 3:
The manuscript investigated the DNA damage toxicity of several metal oxide nanomaterials (ZnO, CuO, TiO2) in alveolar epithelial cells and evaluated their in vitro genotoxicity by high-throughput CometChip assay. The research is less studied and too much discussed, which makes the paper more of a review than a research paper. In addition, I have the following suggestions and questions:
- I wonder that does exposure to MONMs cause it to accumulate in the lungs and cause damage? To be more clear, does MONMS make direct contact with alveolar epithelial cells?
Response:
Acutely, pulmonary exposure to MONMs can result in direct contact between the material and the alveolar epithelial cells. Using enhanced darkfield hyperspectral imaging, our group and others have been able to demonstrate direct interaction between the pulmonary cells with TiO2 (Hussain et al., 2013), CeO2 and other materials such as diesel exhaust particles (Mercer et al., 2018). The propensity of an MONM to accumulate in the lung is dependant on its physico-chemical properties such as aspect ratio, solubility, and size. It is well known that some fibres have long residence times within lung tissue and are not easily removed through local physiological mechanisms, resulting in chronic inflammation and damage to the tissue. An insoluble MONM fibre can present a similar hazard. Other mechanisms of toxicity (damage) exist for MONM within the lung, including but not limited to metal ion induced toxicity due to dissolution and metal leaching, oxidative stress, and DNA damage.
References:
Husain, M., Saber, A. T., Guo, C., Jacobsen, N. R., Jensen, K. A., Yauk, C. L., ... & Halappanavar, S. (2013). Pulmonary instillation of low doses of titanium dioxide nanoparticles in mice leads to particle retention and gene expression changes in the absence of inflammation. Toxicology and applied pharmacology, 269(3), 250-262.
Mercer, R. R., Scabilloni, J. F., Wang, L., Battelli, L. A., Antonini, J. M., Roberts, J. R., ... & Hubbs, A. F. (2018). The fate of inhaled nanoparticles: detection and measurement by enhanced dark-field microscopy. Toxicologic pathology, 46(1), 28-46.
- Each paragraph of results should have a corresponding result diagram produced by the author in the manuscript, not in the supplement or have a list from known information, such as 3.1 and 3.3.
Response:
The authors agree that it is important to summarize the important information from each results section into an appropriate figure. However, some figures and tables were moved to Supplementary files, as including all that information in the main manuscript would increase the length of the manuscript without contributing additional information. For example, section 3.1 is supported by substantial characterisation information presented in the form of a Table (Table 1) and incorporating all the electron microscopy images into the main manuscript document was not deemed practical. For section 3.3, the viability information contained in the supplement file (previously Supplementary Figure 5) is now moved to the main manuscript document (labelled as Figure 2 in the revised manuscript)
- It is recommended to segment the ordinate of Figure 2C to show the difference better.
Response:
For Figure 2C, the scaling on the y-axis was made logarithmic so that minute differences in the TiO2 responses can be visualized better. Figure 2 is now referred to as Figure 3 in the revised manuscript.
- It is more meaningful to caption the material property (such as crystal type and Particle size) than to show the Catalogue Number in figuer2.
Response:
Both Figures 2 and 3 have the legend in panel C modified to include a brief summary of the properties of the TiO2 material in question alongside the catalogue number.
- It is suggested to add a discussion about the dose of human exposure as to whether the dose of human exposure can be inferred from the dose of DNA damage limited to cells.
Response:
The authors thank the reviewer for this insightful comment. However, direct extrapolation to the human condition from the immortalized mouse lung epithelial cells here would require the assumption of a number of parameters outside the scope of this publication (ex. Column media height, media density parameters, media viscosity parameters, particle density parameters). Recent work has indicated that while it is possible to conduct such extrapolations on MONMs such as TiO2 (Romeo et al., 2022, Smith and Skinner, 2021), high variability is seen with relation to computation of a human benchmark dose from in vitro data (Romeo et al., 2022). This is thought to be due to the numerous assumptions made by the models as well as differences in particle characteristics for the exposures used for extrapolation. Since this is a developing field, such an approach will be considered for a future manuscript where multiple in vitro exposures may be considered for human extrapolation.
References:
Romeo, Daina, Bernd Nowack, and Peter Wick. "Combined in vitro-in vivo dosimetry enables the extrapolation of in vitro doses to human exposure levels: A proof of concept based on a meta-analysis of in vitro and in vivo titanium dioxide toxicity data." NanoImpact 25 (2022): 100376.
Smith, J. N., & Skinner, A. W. (2021). Translating nanoparticle dosimetry from conventional in vitro systems to occupational inhalation exposures. Journal of Aerosol Science, 155, 105771.
- Since it is a high-throughput in vitroCometChip assay, why not add the corresponding cometassay pictures in the manuscript?
Response:
Corresponding representative fluorescent comet micrographs are available in Supplementary Figure 5 - 6. As the comet images themselves do not add much value to the main manuscript document, they were placed in the supplement as supporting information.

Reviewer 4 Report
the manuscript is fine, but I am missing two crucial things-first is the comet assay chip protocol and the other is explanation what some of the nanomaterials were characterized in dried conditions and not in the complete medium. references are not complete

Author Response
Reviewer 4:
(Comments extracted from marked PDF)
- I think it is only h or hours (Page 1)
Response:
The manuscript was amended throughout (and all associated figures) to “h”
- Remove bold characteristic of letters, in vitro is in italic (Page 1 keywords)
Response: revised.
- For all chemicals and equipment manufacturer, city, state (Page 3 culture conditions)
Response:
Information pertaining to all chemicals and equipment (manufacturer, city, country) have been added to all relevant methods sections.
- Why only dried samples? you probably already have pictures from sigma aldrich, why you did not make pictures in the medium with serum, where it is known that they can make agglomerates (Page 3; particle size determination)
Response:
The purpose of the EM imaging was to determine the primary particle size analysis (with respect to the MONMs) and to examine the morphology of the MPs. It is well known that manufacturer reported particle sizes can vary considerably from lot-to-lot and it is important to conduct supplementary characterization to confirm reported parameters. For particle size characterisation in suspension, DLS was used with respect to the MONMs (Table 1).
- Remove the empty line (page 3 last line)
Response: revised.
- How many g? manufacturer, city, state (Page 4; particle dissolution experiments)
Response:
There is no associated ‘g’ as this is not a centrifugation. It is shaking(100 rpm, rotations per minute) on an orbital shaker. The associated statement was modified as follows:
“ZnO particle suspensions were incubated on an orbital shaker (1 h shaking per day at 100 (rpm) rotations per minute shaking rate) within polypropylene 50 mL conical tubes at 37°C for 0, 24, and 48 h.”
- Manufacturer, city, state (Page 4; particle dissolution experiments; HNO3)
Response: Information pertaining to all chemicals and equipment (manufacturer, country) have been added to all relevant methods sections.
- City,state (Page 4; particle dissolution experiments; ICP OES)
Response: Information pertaining to all chemicals and equipment (manufacturer, country) have been added to all relevant methods sections.
- Write the doses used in the brackets (Page 5; Trypan Blue exclusion assay)
Response:
The doses were written in brackets in the Trypan Blue exclusion assay section:
“Following overnight incubation, the cells were exposed to 1.8 mL of 3 – 5 doses (0 – 108 µg/mL) of MONMs, MPs, and ZnCl2 or CuCl2 as outlined in Supplementary Table 2.”
- Manufacturer, city, state, which trypsin? with edta, what was its concentration? (Page 5; Trypan Blue exclusion assay)
Response:
The details of the trypsin used were clarified (see below):
“Cells were detached from the surface with 0.15 mL of 0.25% Trypsin-EDTA (1X) (Thermofisher Scientific, Whitby, Canada), and resuspended in 0.5 mL of fresh culture medium
- Is there a reference? (Page 5; CometChip assay)
Response:
The manufacturers protocol for the experiment is available from their website, and is provided with the purchase of the CometChip system (https://www.bio-techne.com/p/activity-assays/cometchip-kit_4260-096-k).
- Manufacturer, city, state (Page 5; CometChip assay)
Response: See response to comment 3. Information pertaining to all chemicals and equipment (manufacturer, country) have been added to all relevant methods sections
- With/in? (Page 5; CometChip assay)
Response:
Reviewer’s comment is not clear. However, the methods section on Page 5 – CometChip assay is modified as follows:
“After 30 minutes of equilibration in 100 mL PBS (room temperature; Thermofisher Scientific, Whitby, Canada), the micropatterned agarose CometChip (Cedarlane Labor-atories, Burlington, Canada) was loaded into the macrowell former system, and the re-sidual PBS was aspirated using a VacuSafe system (Integra LifeSciences, Toronto, Canada).”
- And water but not medium were used as psitive control, why not medium? (Page 5; CometChip assay)
Response:
The H2O2 was prepared as a 9.8mM stock in dH2O which was further diluted in cell culture medium to the required exposure concentration. The sentence was modified as follows:
“For each experiment, a 4 h blank medium negative control was used and cells treated with 100 μM H2O2 in cell culture media for 1 h were used as positive assay controls.”.
- Explain what was lysis composed of, manufacturer, city,state (Page 5; CometChip assay)
Response:
The lysis buffer was purchased and thus, its composition is not known. The manufacturer and the country of origin are specified.
- What was the composition of the alkaline denaturation solution? manufacturer,city, state (Page 5; CometChip assay)?
Response:
Requested information is added and the sentence is modified as follows:
“…and acclimatized in alkaline solution (pH ≥ 13, 200 mM NaOH (Sigma Aldrich, Oak-ville, Canada) + 1mM EDTA (Thermofisher Scientific, Whitby, Canada) + 0.1% Triton-X (Fisher Scientific, Whitby, Canada)) for 40 minutes.”
- How many V/cm? why for 50 minutes? (Page 5; CometChip assay)
Response:
The settings for the electrophoresis are stipulated in the manufacturer protocol, and are to be conducted in a specially designed electrophoresis deck (provided with the purchase of a CometChip starter kit) which holds the chips at a defined, fixed level in the solution during the run. The power supply for the deck is supposed to be set to 22V (constant voltage) and 280mA (variable current) for a total of 50 minutes of electrophoresis at 4C. The power supply routinely reaches and maintains 20V at this setting during the 50 minutes of run time. These parameters were optimized by the company during the development of this assay. The V/cm were calculated based on the distance between the anode and cathode, and the relevant methodology section was revised (see below).
“Electrophoresis was carried out in the dark in alkaline conditions (pH ≥ 13, 200 mM NaOH + 1mM EDTA + 0.1% Triton-X), under constant voltage (20 V, 1 V/cm) and varia-ble current (280 mA) for 50 minutes.”
- manufacturer, city, state (Page 5; Cometchip assay SYBR stain)
Response: Information pertaining to all chemicals and equipment (manufacturer, country) have been added to all relevant methods sections
- 2 and 8 is a huge difference number of replicates... (Page 6; Cometchip assay).
Response:
While it is true there is variation in the number of technical replicates in a single experiment, the number of biological replicates (on which the statistical analysis was conducted) were consistent (3 – 4).The reason for the variation in technical replicates is two fold. First, the presence of cells on the surface of the chip following wash can impede the proper location and identification of the underlying comets (when the surface of the chip has many cells remaining, wells have the appearance of a ‘fog’, masking the comets in the agarose that can be scored). Also, optimisation of the technique called for reduction in technical replicates, which had no bearing on the statistical analysis of the results.
- In dried conditions? (3.31 Results particle characterization page 6)
Response:
Yes the sentence is referring to the primary particle size (determined through TEM imaging of dry materials).
- in italic (Page 7; 3.2)
Response: Fixed.
- in italic (page 8)
Response: Fixed.
- in italic (page 8)
Response: Fixed.
- Depended? (page 9)
Response:
The sentence is correct as stated. The response is dose and time-dependant; i.e. the response depends on the dose and the timepoint of assessment.
- Remove empty line (page 9 last line)
Response: The empty line was removed.
- There should be initials for each person and not the full name and surname (page 19; author contributions)
Response:
The contributions section was modified to include the initials of each author in the appropriate section instead of full name and surname:
“Author Contributions: Conceptualization (Toxicological Research), SH; Methodology, AB (toxicology), MLA (dissolution experiments); Formal analysis, AB (toxicology) and MLA (dissolution experiments) and DW (toxicology); Investigation, AB (toxicology) and SASR (toxicology) and MLA (dissolution) and DW (toxicology); Data curation, AB and SASR; Writing—Original draft preparation, AB and SASR; Writing—Review and editing, AB and SH and SASR and MLA (dissolution experiments) and DW; Supervision, SH (toxicology) and PR (dissolution experiments); Project administration, SH and PR; Funding acquisition, SH and PR. All authors have read and agreed to the published version of the manuscript.”
- So there is comet chip protocol, so if you have a reference, put the protocol in supplementary materials (Page 19; from acknowledgements).
Response:
the original protocol (not published) provided to us by Paul White and Julie Cox was for manual CometChips, which was further modified to suit the commercially available CometChips. The protocol used in this manuscript is described in the methods section.

Round 2
Reviewer 1 Report
The paper can be accepted in current form.
Reviewer 2 Report
Manuscript has been corrected according to the recommendations of the reviewer. All suggestions of the reviewer have been analyzed. In conclusion, revised version of the manuscript may be accepted for publication in the journal.
Reviewer 3 Report
My concerns have been addressed in the revised submission. The manuscript is now acceptable.
Reviewer 4 Report
do not have further comments, but i still think that one affiliations written once is enough